# Prognostic Value of N-Terminal Pro-Brain Natriuretic Peptide and High-Sensitivity Troponin T Levels in the Natural History of Transthyretin Amyloid Cardiomyopathy and Their Evolution after Tafamidis Treatment

**DOI:** 10.3390/jcm10214868

**Published:** 2021-10-22

**Authors:** Silvia Oghina, Constant Josse, Mélanie Bézard, Mounira Kharoubi, Marc-Antoine Delbarre, Damien Eyharts, Amira Zaroui, Soulef Guendouz, Arnault Galat, Luc Hittinger, Pascale Fanen, Emmanuel Teiger, Nadir Mouri, François Montestruc, Thibaud Damy

**Affiliations:** 1Assistance Publique-Hôpitaux de Paris (AP-HP), Cardiology Department, Henri Mondor University Hospital, 1 Rue Gustave Eiffel, F-94010 Créteil, France; bezard.melanie@yahoo.com (M.B.); mounira.kharoubi@gmail.com (M.K.); marcantoine.delbarre@gmail.com (M.-A.D.); deyharts@gmail.com (D.E.); amira.zaroui@aphp.fr (A.Z.); soulef.guendouz@aphp.fr (S.G.); arnault.galat@aphp.fr (A.G.); luc.hittinger@aphp.fr (L.H.); emmanuel.teiger@aphp.fr (E.T.); thibaud.damy@gmail.com (T.D.); 2Assistance Publique-Hôpitaux de Paris (AP-HP), French Referral Centre for Cardiac Amyloidosis, Cardiogen Network, Henri Mondor University Hospital, 1 Rue Gustave Eiffel, F-94010 Créteil, France; pascale.fanen@aphp.fr (P.F.); nadir.mouri@aphp.fr (N.M.); 3Assistance Publique-Hôpitaux de Paris (AP-HP), GRC Amyloid Research Institute, Henri Mondor University Hospital, 1 Rue Gustave Eiffel, F-94010 Créteil, France; 4eXYSTAT, F-92240 Malakoff, France; constant.josse@exystat.com (C.J.); francois.montestruc@exystat.com (F.M.); 5Assistance Publique-Hôpitaux de Paris (AP-HP), FHU SENEC, Henri Mondor University Hospital, 1 Rue Gustave Eiffel, F-94010 Créteil, France; 6Assistance Publique-Hôpitaux de Paris (AP-HP), Genetics Department, Henri Mondor University Hospital, 1 Rue Gustave Eiffel, F-94010 Créteil, France; 7Assistance Publique-Hôpitaux de Paris (AP-HP), Biochemistery Department, Henri Mondor University Hospital, 1 Rue Gustave Eiffel, F-94010 Créteil, France

**Keywords:** heart failure, cardiac amyloidosis, transthyretin, biomarkers, prognosis, treatment, tafamidis

## Abstract

Background: We assesse the evolution and prognostic value of N-terminal pro-brain natriuretic peptide (NT-proBNP) and high-sensitivity troponin T (cTnT-HS) in transthyretin amyloid cardiomyopathy (ATTR-CA) before and after tafamidis treatment. Methods and Results: 454 ATTR-CA patients without tafamidis (Cohort A) and 248 ATTR-CA with tafamidis (Cohort B) were enrolled. Event-free survival (EFS) events were death, heart transplant, or acute heart failure. In Cohort A, 27% of patients maintained NT-proBNP < 3000 ng/L and 14% cTnT-HS < 50 ng/L at 12 months relative to baseline levels. In Cohort B, the proportions were 49% and 29%, respectively. In Cohort A, among the 333 patients without an increased NT-proBNP > 50% relative to baseline EFS was extended compared to the 121 patients with an increased NT-proBNP > 50% (HR: 0.75 [0.57; 0.98]; *p* = 0.032). In Cohort A, baseline NT-proBNP > 3000 ng/L and cTnT-HS > 50 ng/L and a relative increase of NT-proBNP > 50% during follow-up were independent prognostic factors of EFS. The slopes of logs NT-proBNP and cTnT-HS increased with time before and stabilized after tafamidis. Conclusion: ATTR-CA patients with increasing NT-proBNP had an increased risk of EFS. Tafamidis stabilize NT-proBNP and cTnT-HS increasing, even if initial NT-proBNP levels were >3000 ng/L. Thus suggesting that all patients, irrespective of baseline NT-proBNP levels, may benefit from tafamidis.

## 1. Introduction

Transthyretin amyloid cardiomyopathy (ATTR-CA) is a systemic life-threatening disease characterized by amyloid fibrils accumulating in the heart [1]. There are two types of ATTR-CA: one hereditary (ATTRv-CA) with mutated genotypes for the protein transthyretin (TTR), and another with wild-type genotypes (ATTRwt-CA) [1].

In the blood, TTR, mainly produced in the liver, transports the retinol-binding protein-vitamin A complex and to a lesser extent thyroxine. However, when TTR tetramer dissociates and the monomers misfolds TTR aggregates resulting in amyloidogenesis. Mutation of TTR (in ATTRv-CA) or aging (in ATTRwt-CA) destabilizers the tetramers and accelerate their dissociation into monomers. The TTR amyloid fibrils formed by the misfolding deposit in several tissues and organs in the body.

Prior to the development of TTR stabilizers, ATTRv was treated by liver transplantation: replacing mutant TTR, in the liver, with wild-type TTR [2]. Currently, several therapeutics are being developed for ATTR-CA, including tafamidis (a TTR stabilizer) [3]. Tafamidis is a benzoxazole carboxylic acid that binds to both wild-type and mutant TTR proteins, at the thyroxine binding site, with high selectivity and affinity [1,2]. This binding kinetically stabilizes the TTR tetramer that substantially slows dissociation, preventing aggregation and depositing of amyloid fibrils: amyloidogenesis.

The phase III, double-blinded, placebo-controlled ATTR-ACT trial (NCT01994889) assessed the benefit of tafamidis in ATTR patients. The all-cause mortality after 30 months was significantly lower in patients treated with tafamidis (30%) compared to those treated with placebo (43%), hazard ratio (HR): 0.70 (95% CI: 0.51–0.96) [4].

N-terminal pro-brain natriuretic peptide (NT-proBNP) and high-sensitivity cardiac troponin T (cTnT-HS) are important diagnostic and prognostic biomarkers used to evaluated cardiac severity in ATTR-CA patients. Increased levels of these biomarkers indicate more severe cardiac involvement and poorer prognosis. NT-proBNP is produced and secreted by myocytes in the left ventricle under pathological conditions [5]. This biomarker, present in blood plasma, is widely used to detect heart failure and cardiac dysfunction [5]. Similarly, cTnT-HS is a biomarker indicative of myocardial injury [6].

In ATTR-CA patients, several biomarkers, including NT-proBNP, cTnT-HS, and estimated glomerular filtration rate (eGFR), are used to grade the severity of disease [7]. In particular, these biomarkers are used in the Grogan and Gillmore staging systems [8,9].

In this study we wanted to assess whether a longitudinal analysis of increases in the NT-proBNP and cTnT-HS levels, would be of prognostic value in ATTR-CA patients. We also wanted to assess whether tafamidis treatment would affect NT-proBNP and/or cTnT-HS levels in these patients. 

## 2. Materials and Methods

### 2.1. Study Population

We conducted an observational database study in consecutive patients diagnosed with ATTR-CA between 2007 and 2020. Patients referred to the French Referral Centre for Cardiac Amyloidosis in Henri Mondor Teaching Hospital for suspected cardiac amyloidosis were enrolled in the ongoing monocentric, longitudinal, observational prospective registry. ATTR was diagnosed either with myocardial fixation on bisphosphonate scintigraphy, without monoclonal gammopathy, or with gammopathy by a positive staining of extra- or endo-myocardial biopsies using Congo red and TTR antibodies. At inclusion, clinical (weight, size, body surface, blood pressure, heart rate, and signs and symptoms of heart failure), electrocardiographic (rhythm, low voltage pattern), echocardiographic (left ventricular ejection fraction and global longitudinal strain), and biological parameters (sodium, potassium, creatinine, cTnT-HS, and NT-proBNP levels) were recorded. The anamnesis was collected from medical files. TTR genotyping was performed to identify patients with wild-type TTR and those harboring TTR variants. Asymptomatic patients diagnosed with ATTRv-CA after family screening were excluded from analysis. Patients gave informed consent for anonymous publication of scientific data. This study complied with the 1975 Declaration of Helsinki and was approved by our local ethics committee (Créteil) and by the French Comité National de l’Informatique et des Libertés (CNIL number 1431858). Data collection was approved by DIRC Ile de France (DC 2009-930).

### 2.2. Selecting Patients for Cohort A and Cohort B

In Cohort A, untreated patients with at least two NT-proBNP assessments before tafamidis treatment were analyzed. In Cohort B, patients required at least one NT-proBNP assessments before, at least one after initiating tafamidis, and one value within 30 days of tafamidis initiation to assess the impact of tafamidis on NT-proBNP levels. Patients in Cohorts A and B with two cTnT-HS assessments, defined as for NTproBNP, were included in the cTnT-HS analyses.

### 2.3. Statistical Analysis

Continuous variables are reported as their mean with the standard deviation (SD) or as median with the interquartile range (IQR), as appropriate. Categorical variables are reported as numbers with associated percentages. The Chi^2^ test or Fisher exact test were used to compare categorical variables. The Wilcoxon rank-sum test was used to compare continuous variables, between different subgroup previously described. The statistical significance was set at an α-risk of 0.05.

Analyses were performed using SAS Software version 9.4 (SAS Institute, Inc., SAS Campus Drive, Cary, NC, USA)

### 2.4. Assessing Evolution of NT-proBNP and cTnT-HS Levels over Time

For both cohorts, changes of NT-pro BNP levels classes with time were explored using the following five categories at 6, 12, and 18 months after first NTproBNP value: NT-pro BNP levels stable ≤3000 ng/L, increase from ≤3000 ng/L to > 3000 ng/L, decrease from >3000 ng/L to ≤3000 ng/L, and stable > 3000 ng/L, as well as in patients who died. Similarly, the changes of cTnT-HS levels with time were explored using the following five categories at 6, 12, and 18 months: cTnT-HS levels stable ≤ 50 ng/L, increase from ≤50 ng/L to >50 ng/L, decrease from >50 ng/L to ≤50 ng/L, and stable > 50 ng/L, as well as in patients who died. Threshold values for classification of NT-proBNP and cTnT-HS values (3000 ng/L and 50 ng/L, respectively) were selected from Grogan staging [8]. These thresholds were validated with our data, thanks to Contal-O’Quigley method used in [8].

For Cohort A, patients were also classified according to whether the initial NT-proBNP level increased with time by 30%, 40%, and 50% relative to the initial level to either increased (30%, 40%, and 50% relative) NT-proBNP or non-increased (30%, 40%, and 50% relative) NT-proBNP groups. Similarly, patients were classified into increased or non-increased cTnT-HS (30%, 40%, and 50% relative) groups.

For Cohort B, all the NT-proBNP levels (log transformed) were plotted. Slopes before and after initiating treatment were evaluated and compared using a two-piecewise regression model for longitudinal data using a fixed change point that corresponded to the initiation of tafamidis. To account for repetitions over time and heterogeneity of the different timepoints across patients, a random model with an Unstructured (UN) Variance Covariance Matrix was used [10].

### 2.5. Assessing the Impact of Changes in NT-proBNP and cTnT-HS Levels on Event-Free Survival

For the analysis of event-free survival (EFS), the events of interest were the first occurrence among death, heart transplant, or acute heart failure. Patients were censored if they initiated any ATTR treatment or if no EFS event (death, heart transplant, or acute heart failure) had occurred before the end of follow-up or the cut-off date (27 September 2020). The Cox proportional hazards regression model was used to calculate HRs. Log-rank tests were used to compare survival curves. The crude and adjusted analyses were performed using the following baseline factors: NYHA class, ATTR variant (ATTRv versus ATTRwt), Gillmore stage [8], and global longitudinal strain (by echocardiography). Baseline is defined as the first NTproBNP value. To identify factors associated with extended EFS, we performed univariate then multivariate analyses using Cox proportional hazard model and associated Wald test. Potential predictors with a *p*-value below 0.05 were selected for the multivariate analysis using backward selection.

## 3. Results

### 3.1. Study Population

Between 2007 and 2020, 2908 patients with suspected ATTR-CA were referred to our Referral Centre and included in the database (Figure 1). Of these, 834 patients were considered eligible for the study with confirmed ATTR-CA. Of the 834 ATTR-CA patients, 454 patients had two or more NT-proBNP assessments before tafamidis treatment or before an EFS event (Cohort A). In the 834 patients with confirmed ATTR-CA, 434 had received tafamidis (Figure 1). Of these, 248 patients had at least a NT-proBNP assessment before and at least one after initiating tafamidis treatment (Cohort B).

### 3.2. Patient Characteristics and Assessment of Changes in NT-proBNP and cTnT-HS Levels over Time in Cohort A

The characteristics of the 454 patients included in Cohort A, are shown in Table 1. Of these, 134 patients (30%) had ATTRv-CA and 320 patients (70%) had ATTRwt-CA. Patients were predominantly men, 374/454 (82%), and the mean age at baseline was 77 ± 9.6 years.

The natural histories of the evolutions of log (NT-proBNP) and log (cTnT-HS) levels during follow up are shown in Figure 2. Untreated ATTR-CA patients are characterized by progressive increase in NT-proBNP and cTnT-HS levels. The slope of NT-proBNP increase is 0.0144 [0.0066; 0.0221] (*p* = 0.0003) and the slope of cTnT-HS increase is 0.0192 [0.0117; 0.0267] (*p* < 0.0001) (Figure 2).

In Cohort A, the following proportions of patients had baseline levels of NT-proBNP maintained ≤3000 ng/L or decreased from >3000 ng/L to ≤3000 ng/L: 49.8% (*n* = 192/385) after 6 months, 30.6% (45/147) after 12 months, and 23.4% (30/128) after 18 months (Figure 3A). At 18 months, 45% (58/128) of the patients were dead. Similarly, the following proportions of patients had baseline levels of cTnT-HS maintained ≤50 ng/L or decreased from >50 ng/L to ≤50 ng/L: 33.1% (97/293) after 6 months, 15.5% (17/110) after 12 months, and 10.8% (10/93) after 18 months (Figure 3B). At 18 months, 62% (58/93) of the patients had died.

### 3.3. Assessment of the Impact of Increased NT-proBNP and cTnT-HS Levels on Event-Free Survival in Cohort A

At the cut-off date and after a median follow-up of 14.2 months (IQR: 3.1–40.4), 234 patients (52%) had reported EFS events (25 deaths and 209 acute heart failure). Among the 209 acute heart failures, 33 patients died after acute heart failure for a total number of deaths of 58 patients. 220 patients have not reported an EFS event: 176 had initiated tafamidis including 28 patients enrolled in the ATTR-ACT study (NCT01994889), and 44 (9.7%) were without an EFS event (death, heart transplant, or acute heart failure). The median EFS was 8.8 months (95% CI: 6.2–14.2) in patients with 50% relative increased NT-proBNP levels (*n* = 121) and 14.5 months (95% CI: 10.9–19.0) in those with non-increased NT-proBNP (*n* = 333), HR: 0.75 (95% CI: 0.57–0.98); *p* = 0.032 (log-rank test) (Figure 4A). The sensitivity analyses using the 30% and 40% relative increased NT-proBNP levels were not statistically significant, the HRs were 0.80 (95% CI: 0.62–1.04; *p* = 0.09) and 0.78 (95% CI: 0.60–1.01; *p* = 0.06), respectively (figure not shown).

We also analysed median EFS in the 308 patients with at least two cTnT-HS assessments, according to whether patients had a 50% relative increased cTnT-HS levels. The median EFS was 12.2 months (95% CI: 5.1–15.4) in patients with 50% relative increased levels (*n* = 48) and 12.6 months (95% CI: 8.3–16.6) in those with non-increased levels (*n* = 260), HR: 0.86 (95% CI: 0.60–1.25); *p* = 0.43 (log-rank test) (Figure 4B). Similarly, the sensitivity analyses using the 30% and 40% relative increased cTnT-HS levels, were not significant with HRs of 0.87 (95% CI: 0.63; 1.21; *p* = 0.40) and 0.91 (95% CI: 0.64–1.28; *p* = 0.58), respectively (figure not shown).

### 3.4. Multivariate Analysis to Identify Factors Associated with Extended EFS in Patients in Cohort A

Univariate analysis identified non-50% relative increased NT-proBNP levels, as well as the baseline classes of NT-proBNP (≤3000 ng/L), cTnT-HS (≤50 ng/L), and eGFR (>45 mL/min/m^2^) to be significantly associated with an extended EFS at a 5% level. Of these, the multivariate analysis, identified non-50% relative increased NT-proBNP levels (HR: 0.66 [95% CI: 0.48–0.90]; *p* < 0.01, Wald test), as well as baseline levels of NT-proBNP ≤ 3000 ng/L and cTnT-HS ≤ 50 ng/L to be significantly and independently associated with extended EFS (Figure 5).

### 3.5. Patient Characteristics and Assessment of the Effect of Tafamidis on NT-proBNP Levels in Cohort B (n = 248)

Among the 248 patients assessed in Cohort B, 63 (25%) had ATTRv and 185 (75%) had ATTRwt (Table 2). The population was mostly males (82%) and with a mean age at baseline of 75.5 ± 9.8 years. The median follow-up from baseline for patients in Cohort B (*n* = 248) was 17.5 months (IQR: 10.9–18.2).

In Cohort B (*n* = 248), 194 patients (78.2%) were still being treated with tafamidis and 54 (21.8%) had discontinued tafamidis. In patients who discontinued tafamidis, 27 patients had died (including 25 before 18-month timepoint), 15 had aggravated disease, 8 had been enrolled in the ATTR-ACT study (NCT01994889), 1 had been enrolled in the APOLLO study (NCT01960348), 1 did not support the treatment, 1 refused to continue treatment, and 1 was non-compliant. 

In Cohort B (*n* = 248) the following proportions of patients had baseline levels of NT-proBNP maintained ≤3000 ng/L or decreased from >3000 ng/L to ≤3000 ng/L: 139/223 (62.3%) after 6 months, 89/162 (54.9%) after 12 months, and 80/145 (55.2%) after 18 months. At 18 months, 17% (25/145) of the patients were dead (Figure 3C).

Similarly, in Cohort B (*n* = 248) the following proportions of patients had baseline levels of cTnT-HS maintained ≤50 ng/L or decreased from >50 ng/L to ≤50 ng/L: 90/204 (44.1%) after 6 months, 48/147 (32.7%) after 12 months, and 57/134 (42.5%) after 18 months. 19% (25/134) of the patients had died (Figure 3D). 

The scatterplots for log (NT-proBNP) before and after tafamidis treatment, are shown in Figure 6A. The slope before initiating tafamidis was 0.0099 [0.0042; 0.0156] compared to 0.0083 [0.0036; 0.0131] after treatment (*p* = 0.70). We also generated the scatterplots for the subpopulations of patients that initiated tafamidis with NT-proBNP levels >3000 ng/L (Figure 6B) and those with levels ≤3000 ng/L (Figure 6C). For patients with NT-proBNP levels >3000 ng/L, the slope before initiating tafamidis was 0.0255 (95% CI: 0.0175–0.0335) compared to 0.0083 (95% CI: −0.0022 to −0.0187) after treatment (*p* = 0.03). Similarly, for patients with NT-proBNP levels ≤3000 ng/L, the slope was 0.0007 (95% CI: −0.0070 to −0.0084) before tafamidis and 0.0080 (95% CI: 0.0025–0.0135) after tafamidis (*p* = 0.15). However, despite a stable positive slope of NT-proBNP levels after tafamidis, in patients with initial NT-proBNP levels ≤3000 ng/L, most of the individual NT-proBNP levels remain within a narrow logarithmic range (8 logs corresponding to 3000 ng/L).

The scatterplots for log (cTnT-HS) levels before and after tafamidis treatment, are shown in Figure 6D. Interestingly, the slope of the linear regression for log (cTnT-HS) before tafamidis, 0.0117 (95% CI: 0.0079 to −0.0155) was significantly reduced after treatment to 0.0029 (95% CI: −0.0015 to −0.0073), *p* < 0.01. In addition, we plotted the log (cTnT-HS) levels of patients with cTnT-HS levels >50 ng/L (Figure 6E) and those with levels ≤50 ng/L (Figure 6F). For patients with cTnT-HS levels >50 ng/L, the slope before initiating tafamidis was 0.0185 (95% CI: 0.0133 to −0.0224) compared to 0.0015 (95% CI: −0.0044 to −0.0074) after treatment (*p* < 0.001). Similarly, for patients with cTnT-HS levels ≤50 ng/L, the slope was 0.0071 (95% CI: 0.0025–0.0118) before tafamidis and 0.0025 (95% CI: −0.0045–0.0094) after tafamidis (*p* = 0.35).

Appendix A shows time intervals of NT-proBNP and cTnT-HS assessments before and after initiating tafamidis in Cohort B.

## 4. Discussion

Our study is the first to describe the natural history of cardiac biomarkers in ATTR and their evolution after tafamidis treatment in a real-life cohort. Our results show that NT-proBNP and cTnT-HS levels increase in ATTR-CA natural history and that 50% relative increases in NT-proBNP levels significantly increases the risk of death, heart transplant, or acute heart failure in ATTR-CA patients. In contrast, increases in cTnT-HS levels were not associated with an increased risk of these events. Baseline levels of NT-proBNP ≤ 3000 ng/L and cTnT-HS ≤ 50 ng/L were found to be independently associated with extended EFS. Furthermore, our results show that tafamidis slows the increasing of NT-proBNP levels associated with amyloidosis evolution in ATTR-CA patients with NT-proBNP levels > 3000 ng/L when tafamidis is initiated. Similarly, tafamidis also slows the increasing of cTnT-HS levels in patients with cTnT-HS levels > 50 ng/L. The proportion of patients who maintained their baseline levels of NT-proBNP ≤ 3000 ng/L or decreased their NT-proBNP levels from >3000 ng/L to ≤3000 ng/L at 12 months was 31% for untreated patients and 55% for those treated with tafamidis.

### 4.1. Cardiac Biomarkers during the Evolution of Transthyretin Amyloid Cardiomyopathy (ATTR-CA)

Amyloid formation is a progressive and dynamic process with cardiac biomarkers reflecting the pathophysiology of the disease. Our results show that cTnT-HS and NT-proBNP levels increase during the natural evolution of ATTR-CA.

NT-proBNP is a cardiac biomarker indicative of cardiac wall stress and levels increase when heart ventricles become overloaded. NT-proBNP levels are used to diagnose congestive heart failure [5,11,12,13,14]. In contrast, cTnT-HS, as a biomarker, specifically detects myocardial damage [15,16]. NT-proBNP is also known to be useful for early detection of cardiac involvement in ATTRv with polyneuropathy and cTnT-HS to assess it severity [17].

In cardiac amyloidosis, fibrillogenesis is a dynamic process. An increase in biomarker levels may be linked to cardiomyopathy evolution, emphasizing the need to slow down their progression. Thus, increasing NT-proBNP levels may be linked with cardiac wall stress associated with amyloid infiltration. While, increasing troponin levels may be linked with the toxicity of amyloid fibrils [18].

### 4.2. Prognosis

The prognostic value of baseline NT-proBNP and cardiac troponin levels in cardiac amyloidosis patients are well established [7,8,9,19,20,21,22,23]. Indeed, the Grogan staging uses NT-proBNP and troponin levels, while the Gillmore staging uses NT-proBNP combined with eGFR to stage ATTR-CA patients [8,9]. Higher biomarker levels are associated with higher Grogan and Gillmore classes and consequently more severe disease. Indeed, a recent study showed that the Gillmore staging system (also referred to as the National Amyloidosis Centre [NAC] ATTR staging) predicted mortality not only at baseline but during the natural evolution of amyloidosis, with a tendency to higher stages as amyloidosis evolves [24]. Furthermore, in a recent study developing a risk score for ATTR-CA patients, QRS duration, cTnT-HS, and NT-proBNP levels were found to be the best predictors of all-cause mortality [25]. Our data are consistent with these previous studies. We found that in our cohort, patients with higher NT-proBNP baseline levels had an increased risk of death, heart transplant, or acute heart failure. This increased risk was not observed for baseline cTnT-HS levels. This may suggest that NT-proBNP levels evolve over time suggesting cardiac involvement. In contrast, cTnT-HS levels seem to indicate more persistent cardiac damage.

### 4.3. Tafamidis

In our analysis of the cohort of patients treated with tafamidis we found that NT-proBNP levels continued to increase after initiating tafamidis but at a constant or reduced rate. We also found that tafamidis had a similar effect on cTnT-HS levels. The activity of tafamidis is reportedly due to its capacity to stabilise the TTR tetramer, reducing the build-up of amyloid fibrils in the heart [1]. This mechanism of action could explain the stabilisation of NT-proBNP levels following tafamidis treatment. Our results suggest a stop or a decrease of amyloid fibrils infiltration in the heart after tafamidis, resulting in stabilisation of NT-proBNP levels. This impact of cardiac amyloidosis treatment on NT-proBNP level has also been observed in light-chain cardiac amyloidosis patients following chemotherapy. Indeed, NT-proBNP levels often decrease with haematological response, often without structural cardiac improvement [18,26]. This might reflect a higher cardiac toxicity of light chains seen in light-chain cardiac amyloidosis compared to TTR amyloid fibrils in ATTR-CA. It could also indicate that tafamidis not only stops myocardial infiltration but also decreases myocardial toxicity, as evidenced by cTnT-HS levels decreasing at a reduced rate with tafamidis. Thus, stabilising serum TTR may lower the rate of amyloid fibrils’ deposition and for some patients effectively reduce cardiac amyloid fibrils deposits. The diminished cardiac infiltration may explain the extended survival observed in patients treated by tafamidis versus placebo in ATTR-ACT trial after 15 months [4].

Interesting, our results correspond with an exploratory endpoint of the ATTR-ACT study showing a slightly higher NT-proBNP levels in patients, at 12 and 30 months (least-squares mean difference, −735.14 [95% CI: −1249.16 to −221.13] at 12 months and −2180.54 [95% CI: −3326.14 to −1034.95] at 30 months, with tafamidis compared to placebo [4]. Overall, our patient baseline characteristics (Cohort B) are comparable with those of patients treated with tafamidis in the ATTR-ACT study, except for baseline NT-proBNP. Indeed, the median NT-proBNP level at baseline, in Cohort B was only 1882.0 ng/L (IQR: 871.0–3513.0) compared with the 2995.9 ng/L (IQR: 1751.5–4861.6) in the tafamidis group in the ATTR-ACT study.

The prolonged EFS for patients with lower biomarker levels before tafamidis treatment in Cohort A, suggests that stabilizing biomarkers at a lower level by initiating treatment early may be optimal. Furthermore, our results also show that patients with higher biomarker levels may benefit from tafamidis treatment. Indeed, in these patients the increases in biomarker levels seem to be reduced with tafamidis.

### 4.4. Study Limitations

Although our data represents a large representative real-life cohort of patients, particularly in this rare disease setting, our study has several limitations. Firstly, we selected our patient populations for analyses based on the availability of NT-proBNP assessments: at least 2 assessments before tafamidis treatment or death (Cohort A) and one assessment before and another after tafamidis (Cohort B). However, a comparison of the patients included and those not included in the cohorts did not show any major imbalance. Secondly, the time intervals between NT-proBNP assessments were not control and varied. Our data were analysed using piecewise regression random model that generates slope models before and after treatment despite this variability in the time intervals. Thirdly, the cTnT-HS levels used for our analyses are different from the troponin assay usually used in cardiac amyloidosis literature. Recently, cTnT-HS assays have replaced the more traditional assays. These new assays detect the same proteins but with a much lower detection limit [27]. It has been reported that the lower cTnT-HS values used to evaluate prognosis differ substantially depending on the assay used [19,28]. Dispenzieri et al., suggest substituting the 54 ng/L value, in the Mayo staging system, by 35 ng/L for the troponin-level obtained by high-sensitivity assay [19]. In our analyses, we do not believe that changes in the value of the cTnT-HS level with different assays would have changed our results. Fourthly, the analyses using cTnT-HS level probably lacked statistical power. Indeed, our patient populations (Cohort A and B) were selected based on the availability of NT-proBNP levels and not cTnT-HS levels. This might explain why 50% increasing of cTnT-HS in Cohort A was not associated with EFS while baseline cTnT-HS was strongly associated with EFS in our multivariate analysis. Thus, substantially fewer patients had less data for these analyses resulting in lack of statistical power. 

Finally, this study was not designed to compare patients treated with tafamidis with those not treated. The patients in the two cohorts were not included during the same time period. Thus, untreated patients from Cohort A that were included in the Cohort B (tafamidis treated patients) may have less severe ATTR-CA. Indeed, a patient included in Cohort A with an EFS event would not have been included in Cohort B. Comparisons of the baseline characteristics of the cohorts support this hypothesis with lower level of NT-proBNP, cTnT-HS, Gillmore stage, and Grogan stage in treated patients. However, the sensitivity analyses that we performed gave results consistent with the results and conclusions herein reported. Nevertheless, this indirect comparison was not an objective of this study. We are currently indirectly comparing these cohorts using an adapted methodology, including the use of propensity scores and matching analysis.

## 5. Conclusions

NT-proBNP and cTnT-HS are biomarkers for cardiac dysfunction that increase as ATTR-CA evolves. Increasing NT-proBNP levels, but not cTnT-HS levels, signal increasing cardiac dysfunction with an increased risk of death, heart transplant, or acute heart failure, in ATTR-CA patients. Our results seem to indicate that tafamidis slows disease evolution in ATTR-CA patients by stabilizing NT-proBNP and cTnT-HS levels.

## Figures and Tables

**Figure 1 jcm-10-04868-f001:**
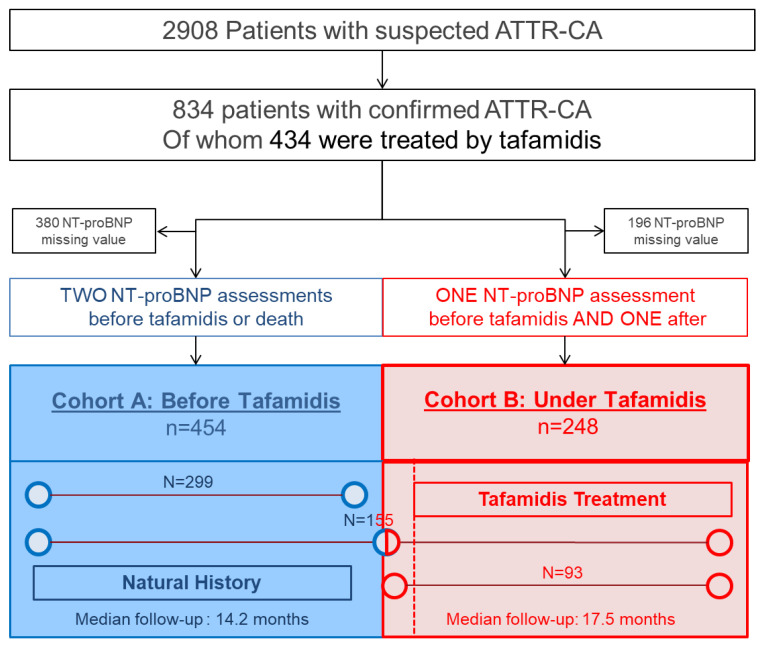
Study flow chart.

**Figure 2 jcm-10-04868-f002:**
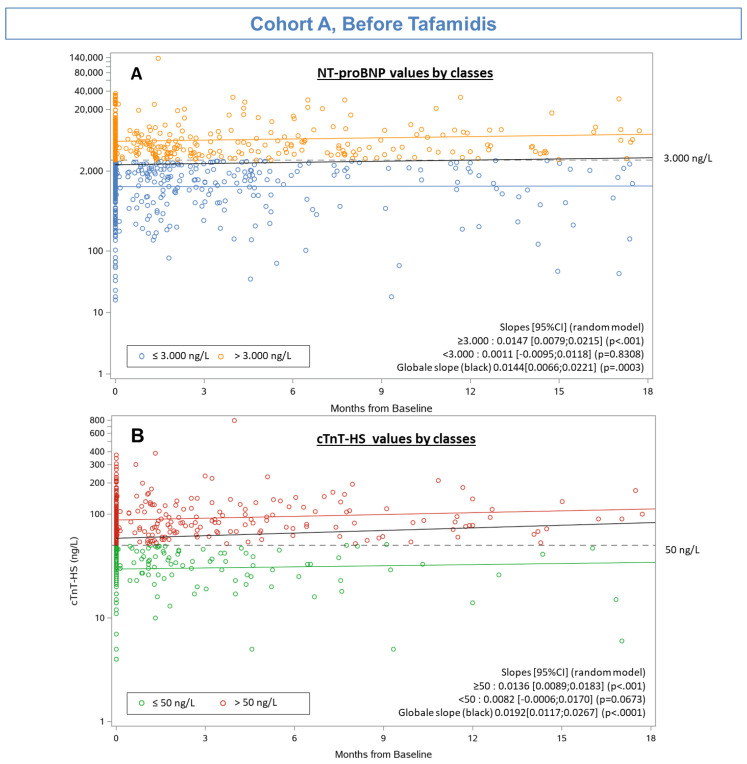
Evolution of biomarkers in Cohort A: Scatter plots of the evolution of log (NT-proBNP) (**A**) and log (cTnT-HS) (**B**) levels with time, for patients in Cohort A (*n* = 454).

**Figure 3 jcm-10-04868-f003:**
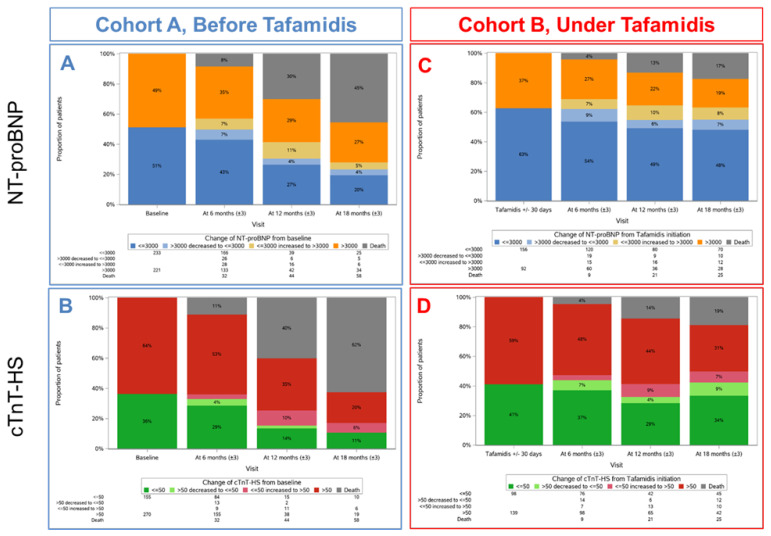
Proportion of patients in Cohort A (*n* = 454) that maintain NT-proBNP (**A**) and cTnT-HS (**B**) levels at 6, 12, and 18 months relative to their baseline levels and proportion of patients in Cohort B (*n* = 248) Proportion of patients in Cohort B (*n* = 248) that maintain NT-proBNP (**C**) cTnT-HS (**D**) levels at 6, 12, and 18 months relative to their baseline levels.

**Figure 4 jcm-10-04868-f004:**
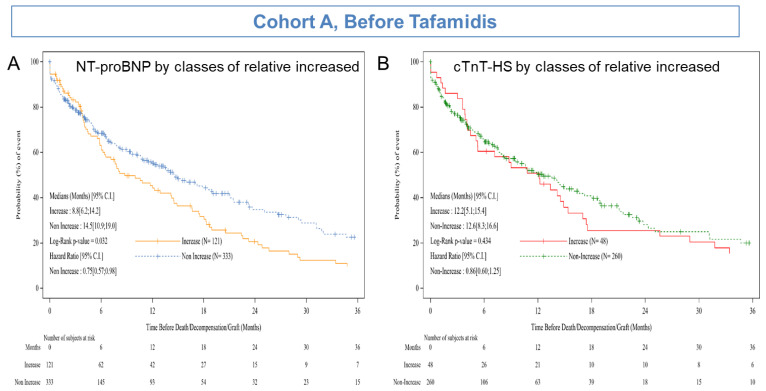
(**A**) Event-free survival by 50% relative evolution of NT-proBNP levels in Cohort A (*n* = 454) and (**B**) Event-free survival by 50% relative evolution of cTnT-HS levels in Cohort A (*n* = 308).

**Figure 5 jcm-10-04868-f005:**
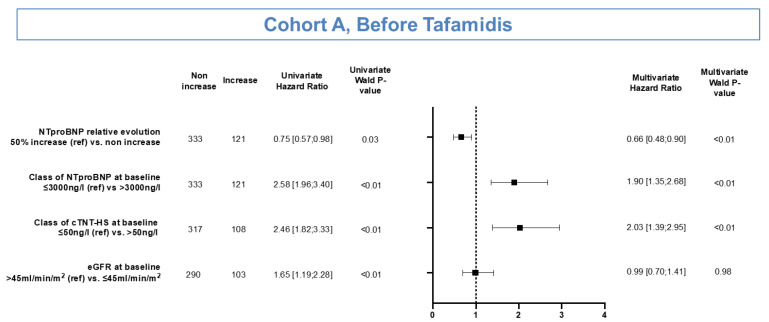
Univariate and multivariate analyses to identify factors associated with extended event-free survival in patients in Cohort A.

**Figure 6 jcm-10-04868-f006:**
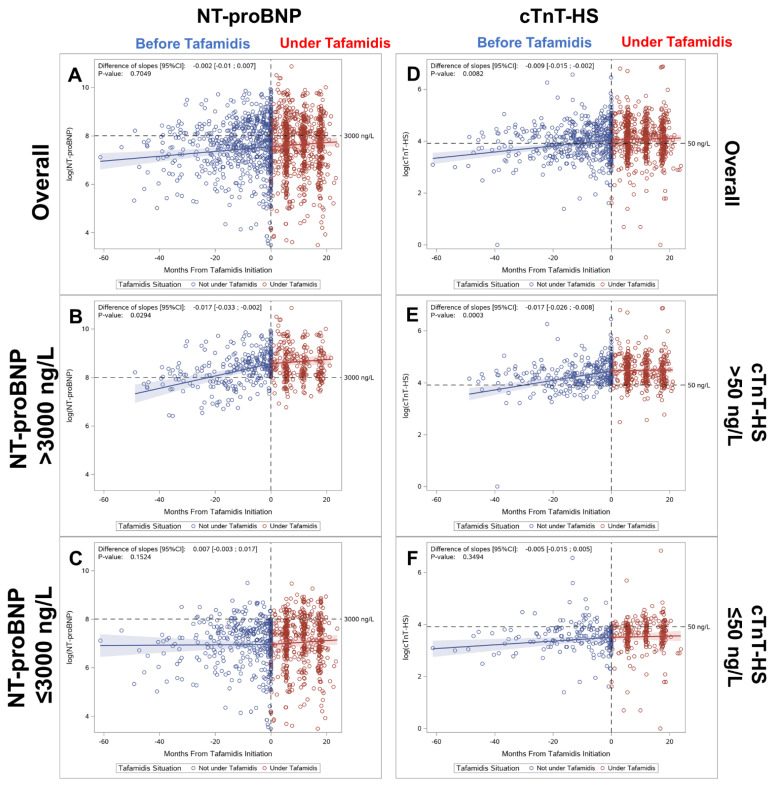
Regression in Cohort B Patients before (blue) and after initiation of tafamidis (red): Of log NT-proBNP (**A**) and depending on NT-proBNP > 3000 ng/L (*n* = 92) (**B**) or NT-proBNP ≤ 3000 ng/L when tafamidis was initiated (*n* = 156) (**C**); Regression of log cTnT-HS (**D**) and depending on cTnT-HS > 50 ng/L when tafamidis was initiated (*n* = 139) (**E**); or with cTnT-HS ≤ 50 ng/L (*n* = 98) (**F**).

**Table 1 jcm-10-04868-t001:** Characteristics of patients (Cohort A), according to genotype, assessed for the impact of NT-proBNP levels on event-free survival (*n* = 454).

	Cohort A	ATTRv-CA	ATTRwt-CA	*p*-Value
*n*	454	134	320	
Age at inclusion, years	77.0 ± 9.6	70.1 ± 11.2	79.8 ± 7.1	<0.01 (ANOVA)
Age at amyloidosis diagnosis, years	76.9 ± 9.6	70.0 ± 11.4	79.8 ± 7.1	<0.01 (ANOVA)
Male, *n* (%)	374 (82.4)	96 (71.6)	278 (86.9)	<0.01 (Chi^2^)
BMI, kg.m^−2^	25.3 ± 3.7	24.8 ± 3.7	25.4 ± 3.7	0.16 (Chi^2^)
Carpal tunnel history, *n* (%)	319 (70.3)	98 (73.1)	221 (69.1)	0.39 (Chi^2^)
ATTR V122I, *n* (%)	89 (19.6)	89 (66.4)	0 (0.0)	NA
ATTR V30M, *n* (%)	13 (2.9)	13 (9.7)	0 (0.0)	NA
NYHA class, *n* (%)				0.43 (Chi^2^)
I	45 (13.4)	15 (16.3)	30 (12.3)	
II	161 (48.1)	38 (41.3)	123 (50.6)	
III	112 (33.4)	33 (35.9)	79 (32.5)	
IV	17 (5.1)	6 (6.5)	11 (4.5)	
Heart rate, bpm	74.8 ± 15.8	75.2 ± 15.6	74.6 ± 15.9	0.74 (ANOVA)
Systolic blood pressure, mmHg	128.0 ± 20.9	120.6 ± 19.8	130.7 ± 20.7	<0.01 (ANOVA)
Diastolic blood pressure, mmHg	74.6 ± 12.4	72.9 ± 10.0	75.2 ± 13.2	0.12 (ANOVA)
Atrial fibrillation, *n* (%)	102 (22.5)	11 (8.2)	91 (28.4)	<0.01 (Chi^2^)
NT-proBNP at baseline, ng/L *	2818(1460; 5571)	2463(839; 5278)	3216 (1701; 5623)	0.07 (ANOVA)
Time interval between NT-proBNP values, months, median (IQR)	4.8 (1.7–12.9)	5.0 (1.7–13.4)	4.6 (1.7–12.8)	0.75 (ANOVA)
Class of NT-proBNP at baseline				<0.01 (Chi^2^)
≤3000 ng/L	233 (51.3)	83 (61.9)	150 (46.9)	
>3000 ng/L	221 (48.7)	51 (38.1)	170 (53.1)	
cTnT-HS at baseline, ng/L *	62.0 (41.0; 91.0)	62.0 (37.0; 92.0)	62.5 (42.0; 90.5)	0.77 (ANOVA)
Time interval between cTnT-HSvalues, months, median (IQR)	3.8 (1.5–11.4)	4.0 (1.4–10.0)	3.7 (1.5–11.8)	0.52 (ANOVA)
Class of cTnT-HS at baseline				0.68 (Chi^2^)
≤50 ng/L	155 (36.5)	46 (38.0)	109 (35.9)	
>50 ng/L	270 (63.5)	75 (62.0)	195 (64.1)	
eGFR at baseline, mL/min/1.73 m^2^ *	59.2 (46.9; 76.5)	59.1 (43.1; 82.2)	59.2 (49.0; 74.7)	0.15 (ANOVA)
Gillmore stage at baseline				0.02 (Chi^2^)
Stage 1	192 (48.9)	72 (55.0)	120 (45.8)	
Stage 2	137 (34.9)	33 (25.2)	104 (39.7)	
Stage 3	64 (16.3)	26 (19.8)	38 (14.5)	
Missing data	61	3	58	
Martha Grogan stage at baseline				0.26 (Chi^2^)
Stage 1	126 (29.6)	42 (34.7)	84 (27.6)	
Stage 2	121 (28.5)	35 (28.9)	86 (28.3)	
Stage 3	178 (41.9)	44 (36.4)	134 (44.1)	
Missing data	29	13	16	
LVEF, %	48.5 ± 12.3	50.7 ± 12.2	47.4 ± 12.3	0.18 (ANOVA)
IVST, mm	17.8 ± 3.4	17.9 ± 3.9	17.7 ± 3.2	0.77 (ANOVA)
LV Global strain, %	−10.8 ± 3.4	−10.9 ± 3.6	−10.8 ± 3.3	0.90 (ANOVA)

Data are presented as mean ± SD or *: median (IQR) or *n* (%) when specified. ATTRv, variant transthyretin amyloidosis; ATTRwt, wild-type transthyretin amyloidosis; BMI, body mass index; cTnT-HS, high-sensitivity troponin T; eGFR, estimated glomerular filtration rate; IQR, interquartile range; IVST, interventricular septal thickness; LV, left ventricular; LVEF, left ventricular ejection fraction; NT-proBNP, *N*-terminal pro-brain natriuretic peptide; NYHA, New York Health Association; SD, standard deviation.

**Table 2 jcm-10-04868-t002:** Characteristics of patients treated with tafamidis (Cohort B) and with NT-proBNP assessments before and after initiating tafamidis (*n* = 248).

	Cohort B	ATTRv-CA	ATTRwt-CA	*p*-Value
N	248	63	185	
Age at inclusion, mean ± SD	75.5 ± 9.8	67.5 ± 11.7	78.2 ± 7.3	<0.01 (ANOVA)
Age at amyloidosis diagnosis, mean ± SD	75.4 ± 9.8	67.3 ± 11.7	78.1 ± 7.3	<0.01 (ANOVA)
Male, *n* (%)	203 (81.9)	44 (69.8)	159 (85.9)	<0.01 (Chi^2^)
BMI(kg/m^2^), mean ± SD	25.2 ± 3.3	24.5 ± 3.1	25.4 ± 3.4	<0.01 (ANOVA)
Carpal tunnel surgery/symptoms, *n* (%)	193 (77.8)	47 (74.6)	146 (78.9)	0.48
ATTR V122I, *n* (%)	33 (13.3)	33 (52.4)	0 (0.0)	
ATTR V30M, *n* (%)	12 (4.8)	12 (19.0)	0 (0.0)	
NYHA class, *n* (%)				0.82 (Chi^2^)
I	27 (14.3)	6 (15.0)	21 (14.1)	
II	104 (55.0)	20 (50.0)	84 (56.4)	
III	57 (30.2)	14 (35.0)	43 (28.9)	
IV	1 (0.5)	0 (0.0)	1 (0.7)	
Heart rate (bpm), mean ± SD	75.2 ± 14.1	78.9 ± 16.6	74.1 ± 13.1	0.05 (ANOVA)
Systolic blood pressure, mean ± SD	132.2 ± 20.5	125.9 ± 22.1	134.0 ± 19.8	0.02 (ANOVA)
Diastolic blood pressure, mean ± SD	75.5 ± 13.2	74.7 ± 12.2	75.7 ± 13.5	0.67 (ANOVA)
Atrial fibrillation, *n* (%)	49 (20.5)	6 (9.8)	43 (24.2)	0.02 (Chi^2^)
NT-proBNP at baseline (ng/L), median (IQR)	1980 (1008; 3810)	1034 (341; 2546)	2166 (1249; 4005)	<0.01 (ANOVA)
NT-proBNP class, *n* (%)				0.02 (Chi^2^)
≤3000 ng/L	168 (67.7)	50 (79.4)	118 (63.8)	
>3000 ng/L	80 (32.3)	13 (20.6)	67 (36.2)	
cTnT-HS at baseline, median (IQR)	55.0 (32.0; 75.0)	51.5 (25.0; 73.0)	56.0 (34.0; 75.0)	0.26 (ANOVA)
cTnT-HS class, *n* (%)				0.74 (Chi^2^)
≤50 ng/L	108 (46.2)	26 (48.1)	82 (45.6)	
>50 ng/L	126 (53.8)	28 (51.9)	98 (54.4)	
Missing data	14	9	5	
Creatinine at baseline, mean ± SD	109.0 ± 41.4	102.4 ± 47.5	111.3 ± 39.0	0.15 (ANOVA)
eGFR (mL/min/1.73 m^2^), median (IQR)	62.8 (51.5; 78.7)	68.4 (53.5; 94.7)	61.8 (49.9; 76.4)	<0.01 (ANOVA)
Gillmore stage, *n* (%)				0.18 (Chi^2^)
Stage 1	148 (61.4)	44 (71.0)	104 (58.1)	
Stage 2	68 (28.2)	14 (22.6)	54 (30.2)	
Stage 3	25 (10.4)	4 (6.5)	21 (11.7)	
Missing data	7	1	6	
Martha Grogan stage, *n* (%)				0.45 (Chi^2^)
Stage 1	98 (41.9)	25 (46.3)	73 (40.6)	
Stage 2	68 (29.1)	17 (31.5)	51 (28.3)	
Stage 3	68 (29.1)	12 (22.2)	56 (31.1)	
Missing data	14	9	5	
LVEF (%), mean ± SD	50.1 ± 12.6	54.9 ± 11.1	48.2 ± 12.8	0.07 (ANOVA)
IVST (mm), mean ± SD	17.7 ± 3.2	17.0 ± 3.6	18.1 ± 3.1	0.27 (ANOVA)
LV Global strain, mean ± SD	−11.0 ± 3.4	−12.9 ± 3.7	−10.2 ± 3.0	<0.01 (ANOVA)

ATTRv, variant transthyretin amyloidosis; ATTRwt, wild-type transthyretin amyloidosis; BMI, body mass index; c-TnT-HS: high-sensitivity troponin T; IQR, interquartile range; IVST, interventricular septal thickness; LVEF, left ventricular ejection fraction; NT-proBNP, *N*-terminal pro-brain natriuretic peptide; NYHA, New York Health Association; SD, standard deviation.

## Data Availability

Not applicable.

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
