# Peer review of "Prognostic Value of N-Terminal Pro-Brain Natriuretic Peptide and High-Sensitivity Troponin T Levels in the Natural History of Transthyretin Amyloid Cardiomyopathy and Their Evolution after Tafamidis Treatment"

_jcm, 2021, doi:10.3390/jcm10214868_

Round 1

Reviewer 1 Report

This is an interesting study evaluating the changes of cardiac biomarkers (NT-proBNP and troponin T) over time in patients with ATTR cardiac amyloidosis, some of them receiving tafamidis. The study has several important limitations acknowledged by the Authors (availability of biomarkers in just a subgroup of patients; use of standard and high-sensitivity assays for troponin T; heterogeneous timing of blood sampling). I may add that the biomarker cut-offs and delta values were arbitrarily selected, and the choice of the composite endpoint (all-cause death, heart transplant or heart failure hospitalization) is questionable. I think that the Authors should search for the most predictive cut-points and consider multiple endpoints. As minor points, NT-proBNP in Table 2 should be expressed as ng/L, and a recent review article dealing specifically with the use of biomarkers for the management of cardiac amyloidosis (doi: 10.1002/ejhf.2113) should be probably cited.

Author Response

Dear Dr Andrès,

Enclosed is the revised manuscript reference number  jcm-1407262, entitled "Prognostic value of N-terminal pro-brain natriuretic peptide and high-sensitivity troponin T levels in the natural history of transthyretin amyloid cardiomyopathy and their evolution after tafamidis treatment". We directly address the concerns expressed by the reviewers and hope you will reconsider our paper for publication in Journal of Clinical Medicine.

We thank the reviewers for their relevant comments, which has helped us further improve our work. A point-by-point reply is provided below.

  • Reviewer comment: The study has several important limitations acknowledged by the Authors (availability of biomarkers in just a subgroup of patients; use of standard and high-sensitivity assays for troponin T; heterogeneous timing of blood sampling).

We thank the reviewer for this comment. We made the choice to include in our study only the 434 patients out of 834 for whom the values of NTproBNP were interpretable given our methodology. However, our cohort is substantial for a publication on a rare disease. It is important to note that the ATTR-ACT data that validated tafamidis was performed in a similar number of patients. Also, this is a “real-life cohort” with all the difficulty of patient follow-up inherent in this type of study.

Regarding our limitation on lacking data for the analysis of cTnT-HS, Table 2 shows that only 14 data are missing for the entire Cohort B. We were probably imprecise in this paragraph since all patients had a cTnT-HS assay. No patient had a standard dosage of troponin. Thus, we have corrected this part, page 14: “Thirdly, the cTnT-HS levels used for our analyses is different from the troponin assay usually used in cardiac amyloidosis literature”. We have also standardized the term used to describe troponin (cTnT-HS) throughout the article to make it easier to understand. We have also corrected the unit of dosage of cTnT-HS and NT-proBNP in the text, tables, and figures (ng/L instead of ng/mL).

Concerning our limitation on the variability of the time intervals, we have added Table 1 Supp showing the variations of the time intervals at each key date (M6, M12, M18) in Cohort B so that  the reader can judge the relevance of the data analyzed. This is a “real-life study” for which the patients could not be summoned precisely at each time point, with precisely 6 months between consultations.

  • Reviewer comment: I may add that the biomarker cut-offs and delta values were arbitrarily selected.

We thank the reviewer for this comment. We added in the Materials and Methods section on page 4 the methodology used for defining these cut-off values: Threshold values for classification of NT-proBNP and cTnT-HS values (3000 ng/L and 50 ng/L, respectively) were selected from Grogan staging.[8] These thresholds were validated with our data, thanks to Contal-O’Quigley method used in [8].

[8] being “Grogan M, Scott CG, Kyle RA, Zeldenrust SR, Gertz MA, Lin G, Klarich KW, Miller WL, Maleszewski JJ, Dispenzieri A. Nat-ural History of Wild-Type Transthyretin Cardiac Amyloidosis and Risk Stratification Using a Novel Staging System. J Am Coll Cardiol 2016;68(10):1014-20.”

  • Reviewer comment: The choice of the composite endpoint (all-cause death, heart transplant or heart failure hospitalization) is questionable. I think that the Authors should search for the most predictive cut-points and consider multiple endpoints.

We chose a composite endpoint frequently used in studies on heart failure. The choice of a composite endpoint associating all-cause death, heart transplant, or acute heart failure seemed relevant to us because it makes it possible to assess the worsening of the disease. By choosing only a single criterion of mortality, we would probably have lacked power, especially since, as indicated in the text, on “234 patients (52%) [who] had reported EFS events 25 [were] deaths and 209 acute heart failure”. Thus, only 10% of events were deaths. Moreover, in the ATTR-ACT study that validated tafamidis in cardiac amyloidosis, a composite criterion was also chosen (mortality from any cause and cardiovascular hospitalization).

  • Reviewer comment: As minor points, NT-proBNP in Table 2 should be expressed as ng/L

We agree with the reviewer and correct this point in tables, figures, and text.

  • Reviewer comment: A recent review article dealing specifically with the use of biomarkers for the management of cardiac amyloidosis (doi: 10.1002/ejhf.2113) should be probably cited.

We thank the reviewer for this comment and have cited this article.

We hope that the new version of our manuscript and our responses have made our article suitable for publication in Journal of Clinical Medicine, and we look forward to receiving your response.

Sincerely, S. Oghina, MD

Reviewer 2 Report

This paper includes interesting material. The authors investigated an important topic in a population with transthyretin amyloid cardiomyopathy. This work describes relatively large population (834 patients with transthyretin amyloid cardiomyopathy), the natural history of NT-pro BNP and cTnT-HS in a real-life cohort, and their evolution after tafamidis treatment.  

Methods:

Please explain why concentration ≤3000 ng/L for NT-pro BNP and 50 ng/L for cTnT-HS  were taken as the cut-off points for the analysis.

References:

Please correct the numbering in the references.

Author Response

Dear Dr Andrès,

Enclosed is the revised manuscript reference number jcm-1407262, entitled "Prognostic value of N-terminal pro-brain natriuretic peptide and high-sensitivity troponin T levels in the natural history of transthyretin amyloid cardiomyopathy and their evolution after tafamidis treatment". We directly address the concerns expressed by the reviewers and hope you will reconsider our paper for publication in Journal of Clinical Medicine.

We thank the reviewers for their relevant comments, which has helped us further improve our work. A point-by-point reply is provided below.

  • Reviewer comment: Please explain why concentration ≤3000 ng/L for NT-pro BNP and 50 ng/L for cTnT-HS were taken as the cut-off points for the analysis.

We thank the reviewer for this comment. We added page 4 in the Materials and Methods section ou methodology for defining these cut-off values: Threshold values for classification of NT-proBNP and cTnT-HS values (3000 and 50 ng/L, respectively) were selected from Grogan staging.[8] These thresholds were validated with our data, thanks to Contal-O’Quigley method used in [8].

[8] being “Grogan M, Scott CG, Kyle RA, Zeldenrust SR, Gertz MA, Lin G, Klarich KW, Miller WL, Maleszewski JJ, Dispenzieri A. Nat-ural History of Wild-Type Transthyretin Cardiac Amyloidosis and Risk Stratification Using a Novel Staging System. J Am Coll Cardiol 2016;68(10):1014-20.”

  • Reviewer comment: Please correct the numbering in the references.

We thank the reviewer for this comment. We have corrected the references.

We hope that the new version of our manuscript and our responses have made our article suitable for publication in Journal of Clinical Medicine, and we look forward to receiving your response.

Sincerely, S. Oghina, MD

Round 2

Reviewer 1 Report

The Authors have modified their manuscript according to my suggestions. I have no further comments.